# Role of Innate Immune Cells in Psoriasis

**DOI:** 10.3390/ijms21186604

**Published:** 2020-09-09

**Authors:** Yuki Sato, Eisaku Ogawa, Ryuhei Okuyama

**Affiliations:** Department of Dermatology, Shinshu University School of Medicine, 3-1-1 Asahi, Matsumoto 390-8621, Japan; yukisato@shinshu-u.ac.jp (Y.S.); ukasie@shinshu-u.ac.jp (E.O.)

**Keywords:** psoriasis, innate immunity, γδ T cell, NKT cell, NK cell, innate lymphoid cells

## Abstract

Psoriasis is a chronic inflammatory skin condition caused by a combination of hereditary and environmental factors. Its development is closely related to the adaptive immune response. T helper 17 cells are major IL-17-producing cells, a function that plays an important role in the pathogenesis of psoriasis. However, recent findings have demonstrated that innate immune cells also contribute to the development of psoriasis. Innate lymphoid cells, γδ T cells, natural killer T cells, and natural killer cells are activated in psoriasis, contributing to disease pathology through IL-17-dependent and -independent mechanisms. The present review provides an overview of recent findings, demonstrating a role for innate immunity in psoriasis.

## 1. Introduction

Psoriasis is a chronic inflammatory skin disease characterized by unique skin symptoms, most commonly manifesting as erythema covered by silvery lamellar scales. Although its etiology remains incompletely understood, recent studies have revealed that pathological crosstalk between immune cells and keratinocytes, which is activated by a combination of hereditary and environmental factors, drives the development and progression of psoriasis.

In 1979, cyclosporine A was identified to be an effective psoriasis treatment [1], suggesting the involvement of T-cell immunity in psoriasis because cyclosporine A suppresses T-cell activity via the calcineurin phosphatase pathway. Subsequently, the role of CD4^+^ T cells in psoriasis has been intensely investigated, revealing that psoriasis-like dermatitis develops in mice transplanted with CD4^+^ T cells from psoriasis patients [2,3]. Furthermore, cytokines secreted by T helper 1 (Th1) cells, including interferon-γ (IFN-γ) and tumor necrosis factor-α (TNF-α), are increased in the psoriatic lesions and peripheral blood of psoriasis patients [4,5,6]. Therefore, Th1 cells have been the predominant focus of psoriasis research. However, administration of IFN-γ and TNF-α does not reproduce psoriasis pathology [7,8,9], suggesting that Th1 cell induction of experimental psoriasis cannot be fully explained by Th1 cytokine function.

In 1996, IL-17 was newly identified as a proinflammatory cytokine produced by human CD4^+^ T cells [10]. IL-17 is comprised of six subtypes, from IL-17A to IL-17F [11,12]. IL-17A, which induces cell proliferation, abnormal cell differentiation, and production of cytokines, chemokines, and antimicrobial peptides in keratinocytes, is increased in psoriatic lesions [11,12]. IL-17-producing CD4^+^ T cells, T helper 17 cells (Th17), were found to play a role in psoriasis [13,14]. Furthermore, IL-23 is essential to the maintenance of Th17 cells, and is abundant in psoriatic lesions [15,16,17]. The IL-23/IL-17 cascade is thought to play a major role in psoriasis, particularly because clinical outcomes of psoriasis are dramatically improved by therapies targeting IL-17 and IL-23 [18,19,20]. Th17 cells have been regarded as a major source of IL-17.

However, IL-17 is also produced by other immune cells involved in the innate immune response, including group 3 innate lymphoid cells (ILC3s), γδ T cells, natural killer (NK) cells, and natural killer T (NKT) cells [21,22]. Similar to Th17 cells that regulate adaptive immunity, these innate immune cells appear to contribute to the pathogenesis of psoriasis. In the present review, we discuss the newly identified roles of innate immune cells in psoriasis, especially ILC3s, γδ T cells, NK cells, and NKT cells.

## 2. Group 3 Innate Lymphoid Cells (ILC3s)

### 2.1. ILC3 Characteristics

ILCs are a newly identified member of the lymphoid lineage, and are key players in immunomodulated processes such as maintenance of barrier function and early response to infection in tissues that contact exogenous pathogens, such as the skin, lung, and gastrointestinal tract [23]. Unlike T cells and B cells, ILCs do not have an antigen-specific receptor [23,24]. Instead, ILCs are activated through signals from cytokine receptors and NK receptors, and rapidly produce large amounts of cytokines. Based on these characteristics, ILCs are classified as innate immune cells and are roughly divided into three types, ILC1, ILC2, and ILC3, based on cytokine production and transcription factor expression. Similar to the differential cytokine production profiles among different subtypes of effector T cells, ILC1s produce IL-12/IFN-γ, ILC2s produce IL-4/IL-5, and ILC3s produce IL-17/IL-22 [23]. The development of ILC3s requires the transcription factor RORγt. ILC3s are classified into natural cytotoxicity receptor-positive (NCR^+^) ILC3, NCR^–^ ILC3, and lymphoid tissue inducer (LTi) cells [25,26]. NCR^+^ ILC3s produce IL-22 in response to IL-23 and IL-1β stimulation. When activated, NCR^–^ ILC3s produce IL-17, IL-22, and IFN-γ [23]. LTi cells produce lymphotoxin in addition to IL-17 and IL-22 [26].

### 2.2. ILC3s Contribute to the Development of Psoriasis via IL-22

ILC3s are thought to contribute to the pathogenesis of psoriasis because some ILC3s have the ability to produce IL-17 [23]. ILC3s are distributed differentially in normal and psoriatic skin. In normal human skin, a few ILC3s are present, primarily in the upper layer of the dermis, and are absent in the epidermis [27]. While ILC3s are commonly NCR^–^ in normal skin, NCR^+^ ILC3s are increased in psoriatic lesions [28,29] (Figure 1). NCR^–^ ILC3s isolated from healthy skin convert to NCR^+^ ILC3s in response to IL-2, IL-23, and IL-1β stimulation in vitro [25]. NCR^+^ ILC3s isolated from psoriatic lesions produce high levels of IL-22, but not IL-17, under IL-2, IL-23, and IL-1β stimulation in vitro [28] (Table 1). Consistently, IL-22 is closely associated with psoriasis pathogenesis, modulating pathological events such as keratinocyte proliferation and upregulation of antimicrobial molecules such as S100A7, S100A8, and S100A9 [30]. NCR^+^ ILC3s increase IL-22 production by IL-1β and IL-23 secreted from keratinocytes and TNF/nitric oxide-producing dendritic cells, respectively [13,28,31] (Figure 1). Interestingly, ILC3s purified from cultured blood cells induce the formation of psoriasis-like dermatitis in a human skin xenotransplant mouse model, suggesting that ILC3s promote the development of psoriasis independent of other immune cells [32]. Further, ILC3s are located near lymphocytes, not near blood vessels or the epidermis, suggesting that cellular crosstalk between ILC3s and lymphocytes could be important in psoriasis pathogenesis [27].

### 2.3. Peripheral Blood ILC3s in Psoriasis Patients

It is likely that psoriasis pathogenesis involves not only skin ILC3s, but also circulating ILC3s in the peripheral blood. A few ILC3s are present in normal human peripheral blood, most of which are NCR^–^ [28,29,33]. The proportion of NCR^+^ ILC3s to total ILC3s is increased in the peripheral blood of psoriasis patients [28,29], and NCR^+^ ILC3s from psoriasis patients are able to produce IL-22 [28]. Further, the proportion of peripheral blood NCR^+^ ILC3s is positively correlated with skin lesion severity, and decreases in correlation with improvement of psoriatic lesions in response to treatment with an anti-TNF-α antibody [29].

### 2.4. ILC3s in Psoriasis Mouse Models

The involvement of ILC3s in psoriatic pathogenesis has been evaluated using mouse models. Topical application of imiquimod, an immune response modifier, induces psoriasis-like dermatitis accompanied by hyperkeratosis, acanthosis, and inflammatory cell infiltration [34]. The administration of an IL-23 antibody, which blocks a key cytokine in psoriasis, improves imiquimod-induced dermatitis, suggesting that this model has a similar pathological mechanism to psoriasis [35]. Imiquimod does not induce psoriasis-like dermatitis in *Ror^−^*^/−^ mice, suggesting the contribution of ILC3s, γδ T cells, and Th17 cells to imiquimod-induced psoriasis-like dermatitis [35]. Interestingly, imiquimod application induces moderate dermatitis in *Rag1*^–/–^ mice, which lack αβ T cells and γδ T cells, but not ILC3s [36]. In addition, imiquimod does not induce psoriatic dermatitis in *Rag2*^–/–^*/Il2rg*^–/–^ mice, which lack αβ T cells, γδ T cells, and ILCs [35]. Taken together, these findings suggest that ILC3s contribute to the development of imiquimod-induced psoriatic dermatitis.

Intradermal injection of IL-23 also induces psoriasis-like dermatitis [37]. IL-23-injected *Rag1*^–/–^ mice exhibit hyperkeratosis, dilated dermal capillaries, mild acanthosis, and dermal inflammation, similar to IL-23-injected wild-type mice [38]. IL-23 injection induces dermatitis even in the absence of αβ T cells and γδ T cells.

## 3. γδ T Cells

### 3.1. Characteristics of γδ T Cells as Innate Immune Cells

γδ T cells have the properties of innate immune cells. In contrast to classical T cells (αβ T cells), in which the T-cell receptor (TCR) consists of an α and β chain, γδ T cells express TCRs containing a γ and δ chain [39]. Due to decreased numbers of gene fragments, the diversity of γδ chains is much less than that of αβ chains [39,40]. γδ T cells are primarily activated by cytokine stimulation and exhibit effector functions in the absence of TCR stimulation [41]. γδ T cells also express toll-like receptors that recognize intrinsic and extrinsic risk signals [41,42]. Based on these characteristics, which are consistent with those of innate immune cells, γδ T cells are classified as innate immune T cells.

### 3.2. γδ T Cells in Humans

In humans, γδ T cells are classified into Vδ1−Vδ3 by the δ chain [39]. Vδ1 cells are present in barrier tissues such as the skin, and are important in cancer immunity and wound healing [39,43,44]. In the context of cancer immunity, Vδ1 cells in the dermis produce IFN-γ, exert cytotoxic functions similar to those of CD8^+^ T cells [45], and infiltrate into solid tumors such as melanomas [46]. In the context of wound healing, epidermal Vδ1 secrete insulin-like growth factor 1, activating cell proliferation and secretion of other growth factors [47]. Vδ2 and Vδ3 cells are present in the peripheral blood, and Vγ9Vδ2 cells are the major population in human blood [48].

### 3.3. γδ T Cells in Psoriasis Patients

Dermal γδ T cells increase in psoriatic lesions compared with healthy skin, and IL-17-producing γδ T cells also increase in psoriatic lesions [49,50]. Analyses of cultured cells isolated from skin lesions have revealed that the number and proportion of IL-17-producing γδ T cells are higher than those of TCRδ T cells in psoriatic lesions [49]. This finding suggests that γδ T cells contribute to the pathogenesis of psoriasis. IL-1β is secreted by keratinocytes, dendritic cells, and macrophages [51]. Intriguingly, IL-1β expression is higher in psoriatic lesions than in healthy skin, and decreases in parallel with clinical improvement of psoriasis severity [51].

In psoriasis patients, Vγ9Vδ2^+^ T cells decrease in the peripheral blood and increase in psoriatic lesions [52]. Cutaneous lymphocyte antigen (CLA)-positive Vγ9Vδ2^+^ T cells also decrease in the peripheral blood [52]. Intriguingly, the proportion of CLA^+^/Vγ9Vδ2^+^ T cells to total Vγ9Vδ2^+^ γδ T cells increases in disrupted skin with suction blisters, but not in the peripheral blood [52]. CLA^+^ Vγ9Vδ2^+^ T cells could infiltrate from the peripheral blood into inflammatory lesions, such as those of psoriasis. CLA^+^/Vγ9Vδ2^+^ T cells from healthy control subjects produce IFN-γ, TNF-α, and IL-17 in response to in vitro stimulation [52]. Indeed, culture supernatant from CLA^+^/Vγ9Vδ2^+^ T cells activates keratinocytes and induces the expression of intercellular adhesion molecule 1(ICAM-1), human leukocyte antigen (HLA)-DR, IL-6, and TNF-α [52], suggesting pro-inflammatory crosstalk between CLA^+^/Vγ9Vδ2^+^ T cells and keratinocytes.

However, high-throughput screening of TCRs revealed that the proportion of γδ T cells to total T cells is less than 1% in psoriatic skin lesions [53]. The majority of IL-17-producing T-cell clones are αβ T cells, not γδ T cells. It is therefore necessary to further evaluate how γδ T cells are involved in psoriasis, and their relative role in comparison with αβ T cells.

### 3.4. γδ T Cells in Mice

In mice, γδ T cells account for more than 99% of CD3^+^ cells in the epidermis [54]. γδ T cells are classified into Vγ1−Vγ7 by γ chain, and epidermal γδ T cells are predominately Vγ5^+^ [54]. Epidermal γδ T cells are referred to as dendritic epidermal T cells (DETCs) due to their morphology, which is similar to that of dendritic cells (Figure 2). DETCs appear to contribute to the maintenance of skin homeostasis and wound healing [55]. On the other hand, γδ T cells account for about 50% of CD3^+^ cells in the murine dermis [54]. Approximately 40–50% of γδ T cells in the dermis are Vγ4^+^, and about 40% of γδ T cells are Vγ6^+^ [54,56]. Dermal γδ T cells express RORγt, IL-7R, CCR6, IL-23R, and IL-17 [49,57].

### 3.5. γδ T Cells Contribute to Psoriasis-Like Dermatitis in Murine Models via IL-17 Production

γδ T cells have been intensely investigated in mouse models of psoriasis. IL-17-producing γδ T cells increase in the dermis of IL-23-injected mice, which develop psoriasis-like dermatitis [49]. About 90% of IL-17-producing cells in IL-23-induced dermatitis are dermal γδ T cells [49] (Figure 2). In addition, in cultured suspensions of dermal cells stimulated with IL-23, about 90% of IL-17-producing cells are γδ T cells [49]. Together, these findings suggest that γδ T cells are the major source of IL-17 in the IL-23-induced dermatitis mouse model (Table 1).

Furthermore, IL-23 induction of psoriasis-like dermatitis is attenuated in *Tcr^–^*^/–^ mice that lack γδ T cells, as demonstrated by a relatively modest increase in skin thickness and infiltration of inflammatory cells [49]. However, IL-23 induces psoriatic dermatitis in *Tcra^–^*^/–^ mice similarly to wild-type mice despite αβ T-cell depletion [49]. IL-23 induction of IL-17 production is significantly lower in the skin of *Tcrd^–^*^/–^ mice relative to wild-type and *Tcra^–^*^/–^ mice. Taken together, these findings suggest that IL-23 induces psoriatic dermatitis in mice by inducing IL-17 secretion in γδ T cells rather than αβ T cells.

On the other hand, epidermal IL-22-producing γδ T cells increase in IL-23-induced psoriasis-like dermatitis [58]. These γδ T cells express chemokine receptor 6 (CCR6) and do not express Vγ5 (Figure 2). CCR6^+^ γδ T cells could migrate from the dermis to the epidermis to modulate psoriatic dermatitis via IL-22 secretion because Vγ5^-^ cells are rarely present in the normal epidermis.

Similarly, dermal γδ T cells and expression of IL-23 and IL-17 are increased in imiquimod-induced psoriasis-like dermatitis [49]. In this context, IL-23 is primarily secreted by dendritic cells and macrophages. IL-17-producing Vγ4 cells are increased in imiquimod-treated mice, suggesting the contribution of dermal γδ T cells, especially Vγ4 cells, to imiquimod-induced psoriasis-like dermatitis [56] (Figure 2). In addition, imiquimod treatment does not significantly induce psoriatic dermatitis, as indicated by lack of epidermal thickening and inflammatory infiltration, in *Tcrd*^–/–^ mice that lack γδ T cells [49]. Taken together, these findings demonstrate that γδ T cells are essential for the induction of psoriatic dermatitis in imiquimod and IL-23 mouse models.

### 3.6. IL-1β Signaling Is Essential for γδ T-Cell Induction of Psoriasis

While IL-17 production is induced by IL-23 in dermal γδ T cells, IL-17 is more strongly induced by co-stimulation with IL-1β and IL-23 [56] (Figure 2). Consistently, injection of an anti-IL-1β antibody suppresses IL-17 production in γδ T cells of an IL-23 mouse model. Moreover, IL-17 secretion is not induced by IL-23 stimulation of dermal γδ T cells from *Il1*^–/–^ mice in vitro [56]. These findings suggest that IL-1β signaling is essential for γδ T-cell activation and IL-17 production, which is supported by the finding that IL-1β expression is elevated in imiquimod-treated skin [51]. Imiquimod-induced dermatitis is attenuated in *Il1r*^–/–^ mice, as is induction of IL-17 and IL-22 secretion [56]. In addition, in chimeric mice in which *Il1r* is deficient only in γδ T cells, imiquimod treatment does not induce skin acanthosis or expression of IL-17 and TNF-α [51]. On the other hand, IL-22 expression in chimeric mice is comparable to that of wild-type mice [51]. The findings suggest that IL-1β signaling is essential for IL-17 production in dermal γδ T cells in the context of imiquimod-induced psoriatic dermatitis.

Relative to specific pathogen-free mice, germ-free mice exhibit decreased IL-1β, decreased total dermal γδ T cells, and decreased IL-17-producing γδ T cells [51]. IL-23 stimulation of IL-17 production is also decreased in γδ T cells from germ-free mice. This suggests that skin commensal organisms contribute to γδ T-cell activation. Skin commensal organisms are altered in psoriasis [51] and may contribute to the development of psoriasis by regulating γδ T-cell function via IL-1β.

### 3.7. Memory Cell-Like Function of γδ T Cells in Psoriasis

It is likely that γδ T cells have memory cell-like immune function. In the imiquimod model of psoriasis-like dermatitis, imiquimod activates IL-17-producing Vγ4^+^ T cells, inducing the migration of γδ T cells to lymph nodes and subsequent proliferation and long-term survival of γδ T cells [50]. Moreover, a second application of imiquimod results in severe skin inflammation through migration of γδ T cells to the treated area [50]. γδ T cells could possess immunological memory, which could contribute to rapid psoriasis relapse following external imiquimod re-application to psoriasis lesions [59].

## 4. NKT Cells

### 4.1. NKT Cell Characteristics

NKT cells express not only the NK receptor, but also a TCR, which consists of α and β chains. Similar to NK cells, NKT cells ordinarily express CD161 receptor (also known as NK1.1) and the NKG2/CD94 heterodimer (Figure 3). NKT cells are classified into type I and type II according to the TCR type [60]. Type I cells are also known as invariant NKT (iNKT) cells, as these cells use only the single Vα domain of the TCR α chain (mouse, Vα14Jα18; human, Vα24Jα18) and a few Vβ domains of the TCR β chain (mouse, Vβ8.2, β7 or β2; human, Vβ11) [60,61]. This TCR recognizes lipids and glycolipids presented on CD1d, a non-classical major histocompatibility complex class I (MHC-I) in mice [61,62]. Type II NKT cells express other TCR types.

NKT cells rapidly produce large amounts of cytokines in response to various stimuli such as lipids, cytokines, and cellular stressors [61]. Mouse iNKT cells are classified into three types according to cytokine production: NKT1 cells that express T-box transcription factor 21 (T-bet) and produce IFN-γ, NKT2 cells that express promyelocytic leukemia zinc finger (PLZF) and produce IL-4, and NKT17 cells that express RORγt and produce IL-17 [61,63,64] (Figure 3). However, whether human NKT cells can be classified according to cytokine production remains unclear.

### 4.2. NKT Cells in Psoriatic Lesions

NKT cells are found in psoriatic skin lesions [65] and decrease after treatment [66,67]. Furthermore, iNKT cells, specifically Vα24^+^/CD161^+^ iNKT cells, increase in the blood of psoriasis patients [68]. By contrast, other groups have reported decreased Vα24^+^/Vβ11^+^ iNKT cells in the blood of psoriasis patients [69]. Hence, there is no current consensus regarding the change of circulating NKT cells in psoriasis. However, psoriatic dermatitis is induced by transplantation of immune cells from the peripheral blood of psoriasis patients into SCID mice with human skin xenografts, and a proportion of the infiltrating cells express NK receptors [70]. Moreover, psoriatic dermatitis is induced by transplantation of NK receptor-positive cells, including NK cells and NKT cells [71]. Thus, it is likely that NK cells and/or NKT cells contribute to the pathogenesis of psoriasis.

### 4.3. Co-Activation of NKT Cells and Keratinocytes

The association between CD161 expressed on NKT cells and CD1d expressed on keratinocytes in psoriasis has been intensely investigated. In healthy skin, CD1d is expressed only on keratinocytes in the outermost three or four cell layers, extending from the granular layer up to the lipid-rich stratum corneum [65]. On the other hand, in the psoriatic epidermis, CD1d is diffusely expressed on keratinocytes from the basal layer to the outer spinous layer beneath the parakeratotic layers [65]. Interestingly, IFN-γ treatment induces keratinocyte expression of CD1d in vitro [65]. Furthermore, in co-culture with CD1d^+^ keratinocytes pre-treated with IFN-γ, CD94^+^/CD161^+^ NKT cells from psoriasis patients produce large amounts of IFN-γ, which is suppressed by an anti-CD1d antibody [72] (Table 1). This finding suggests that NKT cells produce large quantities of INF-γ by interacting with keratinocyte CD1d, which is in turn upregulated by IFN-γ (Figure 3). NKT cells and keratinocytes may therefore form a positive feedback cell activation loop via IFN-γ.

### 4.4. NKT Cells and IL-17

IL-17 is involved in psoriasis pathology and is produced by some NKT cells, which are termed NKT17 cells [73]. NKT17 cells produce IL-17 in response to stimulation with IL-1β and IL-23 [74] and are present in the lung, liver, marginal lymph nodes, and skin [75] (Figure 3). However, the role of NKT17 cells has not yet been defined in the skin [76,77], and the role of NKT17 cells in psoriasis therefore remains unclear.

Interestingly, mouse studies have demonstrated that CD1-autoreactive T cells contribute to the development of psoriatic dermatitis in hyperlipidemia [78]. The CD1-autoreactive TCR binds to CD1b on dendritic cells, which induces IL-17 production in CD1-autoreactive T cells (Figure 3). CD1 autoreactive T cells are similar to NKT17 cells in terms of IL-17 production in response to invariant TCRs and lipids. Therefore, CD1-autoreactive T cells could also be a source of IL-17 in psoriasis.

## 5. NK Cells

### 5.1. NK Cell Characteristics

NK cells possess both effector and cytotoxic functions, and distinguish infected cells and cancer cells from normal cells using NK receptors [79]. Their effector function is exerted by release of pro-inflammatory cytokines such as IFN-γ [80]. Furthermore, NK cells exert cytotoxicity via secreting perforin, granzymes, and granulysin (GNLY), and induce apoptosis via the Fas ligand [80,81,82]. NK receptors are classified into activating and inhibitory receptors by their effect on NK cell function [83], and NK cells are regulated by the integration of signals from both activating and inhibitory receptors.

According to structure, NK receptors are classified as killer cell immunoglobulin-like receptors (KIRs) and C-type lectin-like receptors (CTLRs) [84,85]. KIRs are classified into inhibitory and activating receptors [83]. Inhibitory KIRs, such as KIR3DL1, generally have long cytoplasmic tails with immunoreceptor tyrosine-based inhibitory motifs. Contrastingly, activating receptors, such as KIR3DS1, generally have short cytoplasmic tails with immunoreceptor tyrosine-based activating motifs [83,86]. CTLRs are also divided into activating and inhibitory types: NKG2A is an inhibitory type; NKG2C, NKG2E, NKG2D, and NKG2F are activating types [85,87].

Many of the genes for NK receptor ligands are encoded in the MHC region. Classical MHC-I receptors, such as HLA-A, -B, and -C in humans, are recognized by KIRs [84]. Among CTLRs, NKG2A and NKG2C bind HLA-E (a non-classical MHC-I), and NKG2D binds MHC class I-related chain (MIC)-A/B (a non-classical MHC-I) and Retinoic acid early transcript (RAET) family receptors, which have homology to MHC-I [81,84,87].

NK cells are roughly divided into two subtypes according to the intensity of CD56 expression [79]. CD56^dim^ NK cells are the major component of NK cells in the peripheral blood, accounting for over 90% of all peripheral blood NK cells, and are therefore referred to as circulating NK cells [88]. CD56^dim^ NK cells express inhibitory receptors such as KIRs, and cytotoxic proteins such as perforin and granzyme B [88]. Contrastingly, CD56^bright^ NK cells are primarily present in peripheral tissues, including the skin, and are a minor component of the peripheral blood [88]. CD56^bright^ NK cells are therefore referred to as tissue-resident NK cells [89]. CD56^bright^ NK cells express NKG2A/CD94 rather than KIRs and cytotoxic proteins, and are capable of producing IFN-γ, TNF-α, and granulocyte macrophage colony-stimulating factor (GM-CSF) [89,90,91].

### 5.2. Effector Function of NK Cells in Psoriatic Lesions

Although the roles of NK cells in psoriasis have been investigated, their contribution to disease development and progression remains incompletely understood. NK cells are increased in psoriatic skin lesions [92,93,94]. CD56^+^/CD3^−^ NK cells account for 5–8% of all infiltrating cells in psoriatic lesions, are comprised primarily of CD56^bright^ NK cells that express the inhibitory receptor NKG2A, and are therefore considered as tissue-resident NK cells [94]. In addition, NK cells isolated from psoriatic lesions express the activation marker CD69, and IL-2 stimulation induces the production of IFN-γ and TNF-α in vitro [94] (Table 1). It is likely that NK cells in psoriatic lesions have effector rather than cytotoxic functions. Moreover, treatment of cultured keratinocytes with media supernatant from activated CD56^+^/CD3^−^ NK cells isolated from psoriatic lesions induces keratinocyte expression of ICAM-1, HLA-DR, C-X-C motif chemokine ligand 10 (CXCL10), C-C motif chemokine ligand 5 (CCL5), and C-C motif chemokine ligand 20 (CCL20), suggesting that NK cell stimulation of keratinocytes leads to intra-epidermal infiltration of inflammatory cells via induction of adhesion molecules and chemokine expression in keratinocytes [94]. In psoriatic lesions, NK cell-induced inflammation may be based on keratinocyte activation via cytokines such as IFN-γ and TNF-α (Figure 4).

Intriguingly, HLA-G, a non-classical MHC-I, is expressed in the psoriatic epidermis, but not in the healthy epidermis [95]. HLA-G is a ligand for KIR2DL4, an activating NK cell receptor; hence, NK cells may be activated by HLA-G expressed on keratinocytes in psoriatic lesions [95,96] (Figure 4).

IL-17, IL-22, and IL-23 are known to play important roles in the pathogenesis of psoriasis, and NK cells could produce both IL-17 and IL-22. IL-17-producing NK cells increase in the synovial fluid of undifferentiated spondyloarthritis patients, a disease closely related to psoriatic arthritis [97]. IL-22-producing NK cells, also referred to as NK-22 cells [98], are present in mucosa-associated lymphoid tissues, such as the tonsils and Peyer’s patches. At this time, the roles of IL-17-producing NK cells and NK-22 cells have not been investigated in psoriasis, and their roles in this context remains to be elucidated, although their pathogenic roles in related diseases are suggestive of a role in psoriasis.

### 5.3. Effector Function of Peripheral Blood NK Cells in Psoriasis Patients

It is not clear whether peripheral blood NK cells contribute to the pathogenesis of psoriasis. The number of peripheral blood NK cells is reduced in psoriasis patients [99,100]. Furthermore, the levels of inflammatory cytokines, including IFN-γ and TNF-α, do not differ between peripheral blood NK cells from psoriasis patients and healthy controls [101]. In addition, in vitro stimulation of IFN-γ and TNF-α production is decreased in NK cells from psoriasis patients relative to NK cells from healthy controls [101].

### 5.4. Cytotoxic Function of NK Cells in Psoriasis

The frequency of GNLY^+^ NK cells is increased in the peripheral blood of psoriasis patients. Similarly, the frequency of psoriatic lesion GNLY^+^ NK cells is increased relative to healthy controls and healthy non-psoriatic skin [102]. Interestingly, genetic analyses suggest a relationship between *GNLY* polymorphisms and psoriasis [103]. On the other hand, a seemingly contrasting report suggested that the expression of Fas ligand in NK cells isolated from the peripheral blood does not differ between psoriasis patients and healthy control subjects [101]. Further studies are expected to resolve these discrepancies.

### 5.5. Genetic Analyses of HLA and KIR in Psoriasis

Polymorphisms in *HLA* genes are associated with psoriasis susceptibility [104,105,106]. The frequencies of *HLA-Cw*0602* and *KIR2DS1* are higher in psoriasis patients than in healthy control subjects [104]. NK cells recognize some types of HLA-C receptors through the activating receptor KIR2DS1 or the inhibitory receptor KIR2DL1 [107]. Furthermore, the frequencies of *HLA-B Bw4-80I* and *KIR3DS1* polymorphisms are higher in psoriasis patients than in healthy control subjects [105]. NK cells recognize HLA-Bw4 through the activating receptor KIR3DS1 or the inhibitory receptor KIR3DL1 [107]. In addition, the frequencies of the *HLA-E*01:01* allele and deletion of *NKG2C*, an activating NK receptor, are also higher in psoriasis patients [106]. Thus, similar to HLA, polymorphisms in activating and inhibitory receptors of NK cells are associated with susceptibility to psoriasis, which may be affected by activation of NK cells through the binding of activating/inhibitory receptors with HLA.

### 5.6. NK Cells in Mouse Models of Psoriasis

In the imiquimod-induced psoriasis-like dermatitis mouse model, NK cells are decreased in the peripheral blood and spleen [108]. However, the role of NK cells in psoriasis mouse models remains unclear, although NK cell maturation markers, such as CD11b, CD43, CD27, and killer cell lectin-like receptor G1 (KLRG1), increase in peripheral blood and spleen NK cells from imiquimod-treated mice [109]. Furthermore, alterations in maturation marker expression are correlated with disease severity [109].

## 6. Conclusions

It is likely that ILC3s, γδ T cells, NKT cells, and NK cells are associated with psoriasis pathogenesis. These cell types are frequently increased in psoriasis lesions. Furthermore, decreased induction of psoriasis in mouse models that lack these cell types and induction of psoriasis by transplantation of these cell types into mouse models indicate that these cells are involved in the development of psoriasis. In addition, peripheral blood NCR^+^ ILC3s decrease after treatment [29], suggesting that these cells are markers of psoriasis severity.

## Figures and Tables

**Figure 1 ijms-21-06604-f001:**
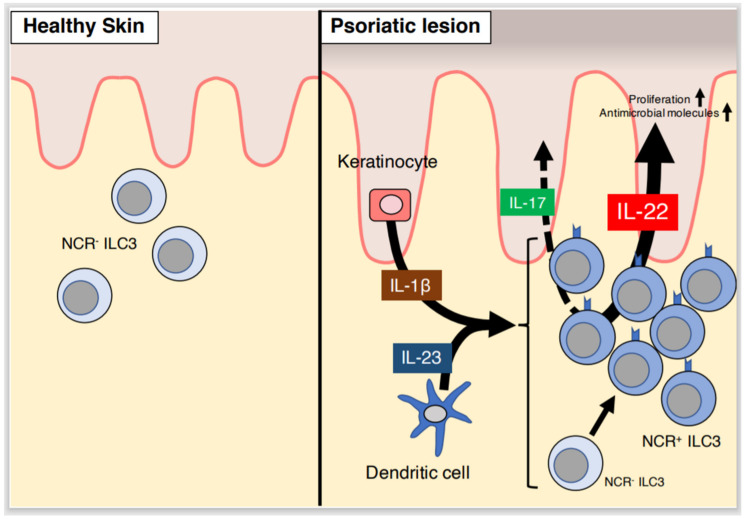
Role of group 3 innate lymphoid cells (ILC3s) in psoriatic lesions. In healthy skin, a few natural cytotoxicity receptor-negative (NCR^–^) ILC3s are present in the dermis. Contrastingly, NCR^+^ ILC3s are increased in psoriatic lesions. NCR^–^ ILC3s may convert to NCR^+^ ILC3s in response to cytokines secreted by keratinocytes and dendritic cells. NCR^+^ ILC3s produce IL-22, rather than IL-17, in response to IL-23 secreted by dendritic cells and IL-1β secreted by keratinocytes. IL-22, and potentially IL-17, secreted by NCR^+^ ILC3s contribute to the pathogenesis of psoriasis.

**Figure 2 ijms-21-06604-f002:**
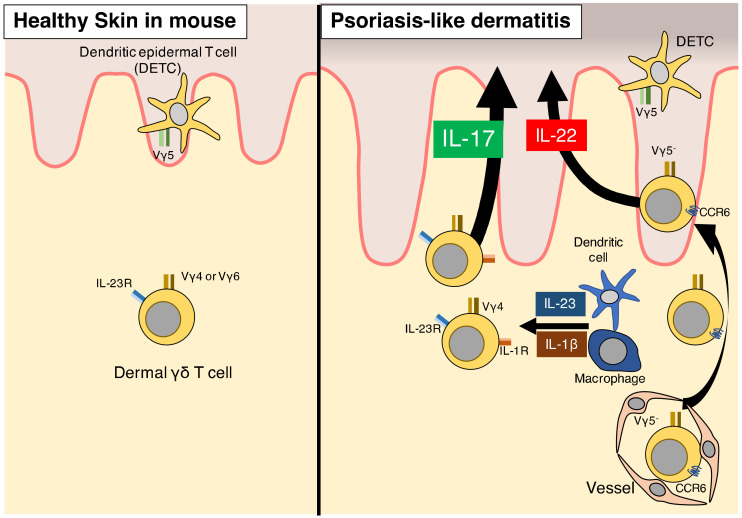
Role of γδ T cells in mouse models of psoriasis. Vγ4 and Vγ6 γδ T cells are present in normal mouse skin. Dermal Vγ4 γδ T cells increase in psoriasis-like dermatitis lesions generated by imiquimod application. Vγ4 and Vγ6 γδ T cells, rather than αβ T cells, produce IL-17. C-C motif chemokine receptor 6-positive (CCR6^+^) γδ T cells lacking Vγ5 are recruited from the peripheral blood into the epidermis and produce IL-22 once recruited. γδ T cells contribute to the development of psoriasis via both IL-17 and IL-22.

**Figure 3 ijms-21-06604-f003:**
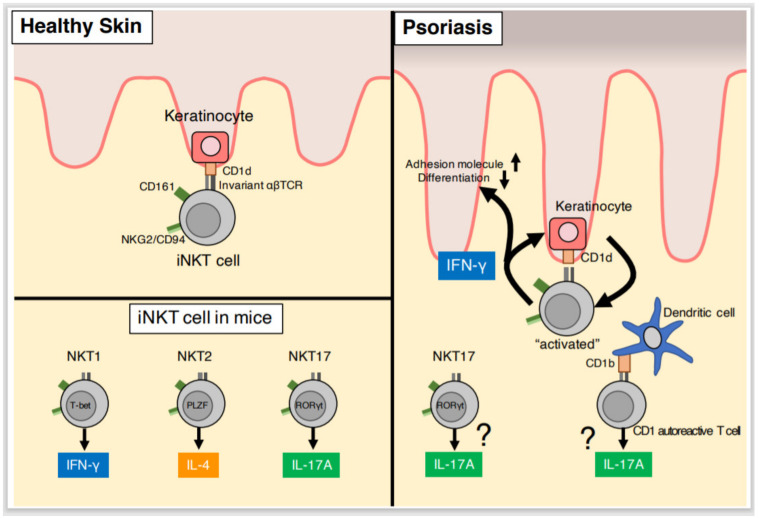
Role of natural killer T (NKT) cells in psoriasis. In psoriatic lesions, CD1d is widely expressed on keratinocytes in all layers of the epidermis. Keratinocyte CD1d expression is induced by interferon-γ (IFN-γ), which is produced by CD94^+^/CD161^+^ NKT cells. The interaction between NKT cells and CD1d expressed on keratinocytes could contribute to psoriasis.

**Figure 4 ijms-21-06604-f004:**
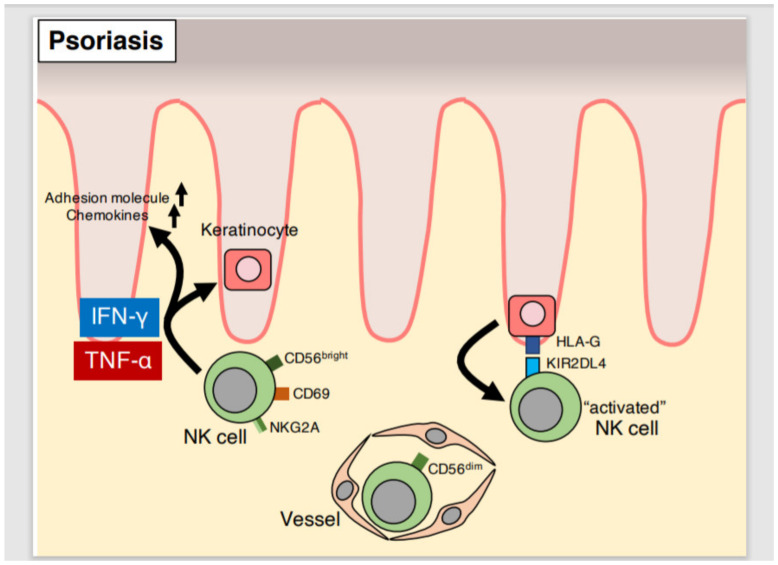
**Role of NK cells in psoriatic lesions.** In healthy skin, CD56^bright^ NK cells are localized to the dermis, and CD56^dim^ NK cells are present predominately in the peripheral blood. In psoriatic skin, CD56^bright^ NK cells are increased and express the activation marker CD69. Keratinocyte HLA-G expression is increased in psoriatic lesions, which may induce NK cell IFN-γ and TNF-α production.

**Table 1 ijms-21-06604-t001:** The Profiles of Innate Immune Cells in the Healthy Condition and Psoriasis.

Cell Type		Tissue	Healthy Condition	Psoriasis
ILC3	Number	Skin	Presence	Increase
		PB	Presence	Increase *
	Phenotype	Skin, PB	NCR^-^	NCR^+^
		Skin, PB		IL-22 (+)
γδT cell	Number	Skin	Presence	Increase
		PB	Presence	Decrease
	Phenotype	Skin		IL-17 (+), IL-22 (+)
NKT cell	Number	Skin	Presence	Increase?
		PB	Presence	Presence
	Phenotype	Skin		IFN-g (+)
NK cell	Number	Skin	Presence	Increase
		PB	Presence	Decrease
	Phenotype	Skin	N.A.	CD69^+^
		Skin		IFN-γ (+), TNF-α (+)

* The number of NCR^+^ ILC3s. N.A., not assessed; PB, peripheral blood.

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
