# Peer review of "Role of Innate Immune Cells in Psoriasis"

_ijms, 2020, doi:10.3390/ijms21186604_

Round 1

Reviewer 1 Report

This manuscript is well written about innate immune cells in psoriasis.

The article contains a broad spectrum of review on innate lymphoid cells, γδ T cells, natural killer T cells, natural killer cells in psoriasis of human and mouse systems.

In figure 2, not Vγ5 is inappropriate. Either Vγ5- or Vγ5neg is recommended.

In 4.4. NKT cells and IL-17 (PDF page 8), please clarify which one is correct between CD1b and CD1d in the text.

Author Response

We really appreciate the Reviewer for insightful comments.

Based on your comments, we have corrected Vg5 to Vg5- in figure 2. In addition, “The CD1-autoreactive TCR binds to CD1b on dendritic cells”, as we described in “4.4. NKT cells and IL-17”, lines 7–8.

Reviewer 2 Report

The title accurately reflect the case report.

The manuscript involves an important aspect of psoriatic inflammation. 

It is well written in terms of clarity, style, and use of English. The manuscript have a logical construction.The figures are relevant and help to understanding.

However I'd like to suggest to create the table with comparison the role of innate cell function in healthy patients and in psoriatic inflammation (summative).

Author Response

We thank the reviewer for helpful and constructive comments.

We have added new table 1 which describing a role of innate cell function in healthy controls and patients with psoriasis.

Reviewer 3 Report

A very up-to-date and detailed review that addresses an
underappreciated part of the immune system in the pathogenesis of
psoriasis.
Especially in the context of cells other than Th17 lymphocytes,
which produce interleukin-17, crucial for the development of
psoriasis.
Figures are a valuable supplement, additionally indicating the interactions between individual cells.
In the "Conclusions", a valuable supplement would be a practical
indication of the role of these cells in psoriasis - perhaps the
influence of the skin dysbiosis, the target for new therapies in psoriasis, changes after traditional antipsoriatic treatment.

Author Response

We thank the reviewer for helpful and constructive comments.

We have added a description of practical indication of the innate immune cells in “6. Conclusion”, lines 5–6.

Reviewer 4 Report

In this manuscript, Sato and colleagues explained the role and function of immune cells in psoriasis with special attention to the innate immune cells. They especially focused on the newly identified roles of ILC3s, γδ T cells, NK cells, and NKT cells, which produce IL-17 and appear to contribute to the pathogenesis of psoriasis, and summarized their characteristics, functions, and roles in both human and mice. Overall, the manuscript is well organized and well written.

I have just minor comments:

- Introduction section, line 6 of 3rd paragraph, “IL-23/IL17” should be “IL-23/IL-17”.

- A space between a word and [reference number] is lost sometimes. The style should be consistent throughout the manuscript. Those are found in 2.2. line 2; 2.4. line 9; 3.1. lines 3 and 5; 3.3. lines 5 and 10; 3.5. line 9; 3.7. line 5; 4.3. line 4; 5.1. line 10 and 21; 5.5. line 1.

Author Response

We thank the reviewer for helpful and constructive comments.

Thank you for your comment. I have corrected the description in “1. Introduction”, line 21.

Furthermore, we have inserted a space where you pointed out. In addition, we have also inserted a space in “2.1. ILC3 characteristics”, line 13.
